# Machine Learning-Based Pattern Recognition of Risk Factors for Low Back Pain among Adolescent Cricket Players in Dhaka City

Marzana Afrooj Ria[1]*, Tasrima Trisha Ratna[2], Shudeshna Chakraborttye Purba[3], Rubal Kar[4], Mohoshina Karim[5], Md Osman Ali[6], Erfat Jaren Chaity[7], Shahadath Hossen[6], Joynal Abedin Imran[1], Shahriar Hasan[8]

**1** Department of Physiotherapy, National Institute of Traumatology & Orthopaedic Rehabilitation (NITOR), Dhaka, Bangladesh, **2** Department of Medicine, Shahabuddin Medical College Hospital, Dhaka, Bangladesh, **3** Department of gastroenterology, Bangladesh medical college, Dhaka, Bangladesh, **4** Department of Physiotherapy, Gono Bishwabidyalay, Savar, Bangladesh, **5** Institute of Nutrition & Food Science, University of Dhaka, Dhaka, Bangladesh, **6** Department of Electrical and Electronic Engineering, Noakhali Science and Technology University, Noakhali, Bangladesh, **7** Department of Pharmacy, East West University, Dhaka, Bangladesh, **8** Department of Public Health, North South University, Dhaka, Bangladesh

* marzana.afrooz@gmail.com

## Abstract

Low back pain (LBP) is common among adolescent cricketers, often due to repetitive lumbar stress. This study investigated LBP among 450 adolescent cricketers in Dhaka City, Bangladesh, considering a range of factors, including sociodemographic characteristics, game-related activities, preventive practices, and LBP-related history. Various ML algorithms applied to LBP severity classification included K-Nearest Neighbors, Random Forest, Logistic Regression, and Support Vector Machine (SVM). LBP severity was categorized into three classes as no pain, mild pain, and moderate pain because there was an insufficient amount of data for the severe pain category. The SVM using the sigmoid kernel of the models considered gave the best performance as it produced the best performance metrics of test accuracy (87.6%), precision (90%), recall (87.6%), and F1-score (87.1%). In addition, regression analysis was also applied to identify the predictors of LBP. Key correlates included female gender (adjusted odds ratio [AOR] = 2.52), higher educational attainment (e.g., undergraduate: AOR = 5.38), elevated family income (e.g., >60,000 BDT: AOR = 4.36), longer weekly practice duration (>20 hours: higher prevalence of 81.7%), inconsistent warm-up (often/sometimes: AOR = 12.48-14.07) and cool-down practices (sometimes: AOR = 2.86), and prior LBP history (AOR = 6.92), all significantly associated with increased LBP risk (p < 0.05). The findings show the importance of early intervention and prevention protocols for minimizing LBP occurrence among junior cricket players. In short, this work demonstrates the effectiveness of ML and regression models for ascertaining sports injury patterns of risk, data-informed prevention and management protocols, and providing a foundation for future studies on this subject. Limitations include the exclusion of a severe pain

**Data availability statement:** All relevant data are within the manuscript and its Supporting Information files.

**Funding:** The author(s) received no specific funding for this work.

**Competing interests:** The authors have declared that no competing interests exist.

category due to insufficient data, which reduces the model's capacity to triage urgent cases requiring immediate intervention.

---

## Author summary

Low back pain (LBP) is a common issue among young cricket players, especially adolescents, due to the repetitive stress on their spines from activities like bowling and batting. In Dhaka, Bangladesh, where cricket is hugely popular, many teens face this problem, but little is known about the specific risk factors in this setting, and tools to predict and prevent it are limited. We surveyed 450 adolescent cricketers aged 11–19 from local clubs, collecting data on their backgrounds, playing habits, and training routines. Using machine learning techniques and statistical analysis, we identified key risks and created a model to classify pain levels as none, mild, or moderate. Our best model achieved about 88% accuracy in predicting pain severity. Factors like being female, higher education or family income levels, longer practice hours, inconsistent warm-ups or cool-downs, and a history of previous LBP significantly increased the risk. These insights highlight the need for better training practices to protect young players. By applying simple data tools, coaches and health workers in low-resource areas like Dhaka can spot at-risk teens early, design personalized prevention plans, and reduce long-term injuries—ultimately helping more kids enjoy cricket safely.

## Introduction

Low back pain (LBP) in adolescent athletes, particularly cricketers, is a growing concern due to repetitive spinal stress during development [1]. Globally, lifetime prevalence of LBP ranges from 35–85% among adolescents, with higher incidence in females [2,3]. In Bangladesh, where cricket is a national passion, studies report high musculoskeletal pain prevalence (~80%) among adolescent cricketers, including lower back commonly affected; however, specific LBP data for adolescent cricketers are scarce, highlighting a critical research gap [4–6]. Under-reporting is prevalent, as adolescents often normalize pain due to cultural factors, limited healthcare access, or fear of impacting sports participation [3,7].

LBP etiology is multifactorial, encompassing biomechanical stresses from cricket-specific movements like hyperextension, rotation, and lateral flexion, which increase risks for immature spines and injuries such as spondylolysis [6,8,9]. Genetic predispositions also play a role, with heritability estimates of 30–50% from twin studies and associations with genes linked to disc degeneration [10,11].

Impacts include reduced physical functioning, emotional well-being, strained relationships, and elevated risks for chronic pain into adulthood [1].

Early detection is essential for effective prevention, enabling personalized training and rehabilitation programs [1]. Traditional manual protocols lack capacity to identify

complex patterns and exhibit poor predictive validity, especially for nonlinear relationships in heterogeneous populations [12,13].

In contrast, machine learning (ML) provides a data-driven approach, processing large datasets on training, biomechanics, and psychosocial factors to stratify risks and forecast LBP probability [14]. Yet, no predictive ML models exist for screening LBP in adolescent cricketers in Bangladesh.

This study addresses this gap using ML and regression analysis to identify key risk factors in Bangladeshi adolescent cricketers. Findings will guide health professionals, coaches, and managers in optimizing prevention and management strategies, serving as a model for ML applications in sports injury research across regions.

## Materials & methods

### Study population

For this study, 450 adolescent cricket players data were collected using purposive sampling technique across various cricket clubs (BKSP, Lt. Sheikh Jamal cricket Academy, Kola Bagan Krira chakra, Khelaghar Cricket Academy and City club) in Dhaka City, with the primary focus on identifying the risk factors associated with the development of LBP. To ensure a balanced dataset for robust machine learning evaluation, this purposive sampling incorporated predefined quotas for each pain severity category. The data collection was conducted from 25 February 2024–28 November 2024. Inclusion criteria included adolescents aged 11–19 years actively participating in cricket training or matches for at least 5 hours per week, with no recent (past 6 months) major injuries unrelated to LBP. Exclusion criteria encompassed individuals with congenital spinal deformities, recent surgical interventions, or unwillingness to participate.

### Ethical consideration

This study adhered to the ethical principles of the Declaration of Helsinki. This study was also approved by the IRB of the Institutional Review Board (IRB) of the National Institute of Traumatology and Orthopaedic Rehabilitation (NITOR/PT/93/IRB/2024/05). Participants were fully informed of the study's purpose, methodology, and procedures. The participants had the right to withdraw from the interview completely at any time. Confidentiality and anonymity were ensured. No physical specimens were collected.

### Outcome variables

Severity was self-reported by participants using the Numeric Pain Rating Scale (NPRS), where pain intensity is rated from 0 (no pain) to 10 (worst imaginable pain) [15]. Categories were defined as: no pain (0), mild (1–3), moderate (4–6), excluding severe (7–10) due to insufficient cases. To facilitate the data collection and classification process, LBP severity was categorized into three distinct classes, such as Class 0: No pain (No LBP), Class 1: Mild pain, Class 2: Moderate pain. Specifically, data collection was prospectively managed and concluded for each category once a predefined quota of 150 valid instances was reached. This resulted in an inherently balanced distribution of exactly 150 participants categorized as Class 0 (No pain), 150 as Class 1 (Mild pain), and 150 as Class 2 (Moderate pain).

### Explanatory variables

The dataset included a variety of features capturing the demographic, physical, and training-related characteristics of adolescent cricket players. Data were collected through structured surveys and physical assessments, and were organized into the following categories:

- Sociodemographic factors: Age, gender, education, monthly family income, and body mass index.

- Games-related factors: Playing position, playing experience, and duration of practice per week.

- Preventive measures factors: Warm-up before sports activity, duration of warm-up, cooldown after sports activity, and duration of cool down.

Players' ages and heights ranged from 11 to 19 years and from 120 to 180.3 cm, respectively. This information provides insight into their physical stature, which may influence biomechanics during cricketing activities. Similarly, players' weight can significantly affect the strain on the musculoskeletal system during high-intensity activities like cricket. In this study, players' weights ranged from 20 to 80 kilograms.

The players' Body Mass Index (BMI) varied between 8.53 and 29.81. This measure was split into three categories according to WHO: underweight (BMI < 18.5), normal (BMI 18.5 - 24.9), and overweight (BMI ≥ 25), with implications for the overall risk of injury to the player [16]. The educational level was divided into four categories: J.S.C./8th grade, S.S.C./O-levels, H.S.C./A-levels, and undergraduate. This variable is important as education may reflect underlying socio-economic factors influencing players' overall health and physical activity.

Position of play was another factor of significance, following categories such as right-hand batsman, left-hand batsman, spin hand bowler, pace hand bowler, wicket-keeper, and all-rounder. These roles dictate the type of biomechanical stress players experience; bowlers particularly pace bowlers, face a higher LBP risk due to repetitive movements [17,18]. The players were classified in terms of their playing experience among three categories, 1–3 years, 4–6 years, and 7–10 years. The players' practice duration per week (hour) was classified in three categories: < 10 hours, 10–20 hours, and >20 hours.

Training habits are captured through the doing and duration of warm-up and cooldown. Warm-up before and cooldown after the sports activity were categorized into 'always', 'often' and 'sometimes'. The warm-up duration ranges from <10 minutes to >15 minutes, while cooldown time is similarly categorized into <10 minutes, 10–15 minutes, and >15 minutes. These factors are critical as appropriate warm-up and cooldown routines are known to reduce the risk of injury and improve muscle recovery. Lastly, past history of LBP is documented either "Present" or "Absent".

In summary, the machine learning model for LBP severity classification was developed using an integrated set of attributes, including sociodemographic factors, game-related factors, preventive, and LBP related factors. The model's performance was evaluated using 10-fold cross-validation. No formal feature selection was applied, as all variables were selected based on domain knowledge and prior literature relevance. Categorical variables were encoded using a hybrid approach in which ordinal and binary features such as Age category, Sex, and Education level were transformed using label encoding to preserve their inherent order, while nominal features were transformed using one-hot encoding to maintain interpretability and ensure compatibility with machine learning algorithms including SVM, as implemented in scikit-learn. This approach offers technical advantages by reducing feature dimensionality, preserving meaningful ordinal relationships, preventing the introduction of spurious ordinal assumptions for nominal variables, and improving both computational efficiency and predictive performance across diverse algorithms.

### Selection of algorithms

The most popular machine learning-based classification models include Logistic Regression (LR), K-Nearest Neighbors (KNN), Random Forest (RF), and Support Vector Machine (SVM), whereas deep learning-based classification models are Convolutional Neural Networks (CNN) and Deep Neural Networks (DNN). These machine learning-based model were chosen in this study because of their effectiveness in handling structured data in contexts with a small number of samples. The deep learning models are in need of a large volume of data to automatically extract and learn intricate feature representation efficiently [19].

This study evaluates the performance of four machine learning classifiers: SVM, RF, KNN, and LR for the classification of LBP severity among adolescent cricket players. The dataset was balanced, consisting of 150 instances in each of the three classes: No pain, Mild pain, and Moderate pain. Of the above-mentioned machine learning algorithms, KNN is a

non-parametric model whereas RF and SVM have good classification properties and are suitable for non-linear decision boundary modelling.

## Logistic Regression

LR is a widely used statistical method for binary and multiclass classification in medical research [20]. It models the probability of class membership using the logistic function, which is a type of sigmoid function. This function assumes a linear relationship between the independent variables and the log-odds of the outcome. It is efficient, interpretable, and works best when the data has a linear decision boundary. It is sensitive to multicollinearity and may underperform in complex or nonlinear datasets unless extended with interaction terms or regularization [21]. In this study, logistic regression served as a baseline model. While in this study, its simplicity allowed clear insights into individual predictors of low back pain, it was less effective in capturing nonlinear associations compared to tree-based or kernel-based models.

## K-Nearest neighbours

KNN is a non-parametric, instance-based learning algorithm commonly used in both classification and regression tasks, particularly within structured medical data contexts due to its simplicity and effectiveness in handling health-related patterns [22]. It predicts outcomes by identifying the $k$ closest data points to a given instance based on a distance metric, typically Euclidean distance:

$$Distance = \sqrt{(x_2 - x_1)^2 + (y_2 - y_1)^2}$$

KNN performs well on small to moderately sized datasets with low to medium dimensionality and balanced class distributions. It is particularly useful when the data exhibits local patterns or clustering. However, its performance can degrade in high-dimensional or noisy datasets, and it becomes computationally expensive with large datasets due to its lack of a training phase.

In this study, KNN was applied during the model evaluation phase to explore pattern similarity among adolescent cricket players.

## Random forest

Random Forest (RF) is an ensemble learning algorithm [23] that builds multiple decision trees and combines their outputs to improve classification accuracy and reduce overfitting. Each tree is trained on a random subset of the data and features, and final predictions are made by majority voting for classification or averaging for regression. It is well-suited for handling datasets with nonlinear relationships, mixed data types, and moderate levels of noise. It provides good performance even without extensive parameter tuning and offers feature importance scores, aiding interpretability. However, its complexity can increase with many trees, leading to slower predictions and less transparency compared to simpler models.

In this study, Random Forest demonstrated strong classification performance in identifying low back pain patterns, particularly due to its robustness against overfitting and ability to model interactions between risk factors.

## Support Vector Machine (SVM)

SVM is a supervised learning machine learning algorithm with excellent performance in multiclass classification even when it has been trained on relatively small data sets. It is a human-interpretable model and has good generalizing capacity on small data samples and is less susceptible to overfitting in comparison to ensembled-based models such as RF [24–26]. SVM generates optimal separating decision boundaries, called hyperplanes, in a high-dimensional feature space to classify data into their

respective classes. These hyperplanes are maximally positioned between classes. In other words, the hyperplanes are put in such a way that they are maximally distant from data points of any class. These data points are called support vectors. Maximizing this distance between classes is what aims to help improve the model's performance in generalizing new data [24].

Several mathematical functions known as kernels are employed to deal with non-linearly separable data like linear kernel, polynomial kernel, radial basis function (RBF), and sigmoid. These kernels enable the mapping of original data to a higher-dimensional feature space, where it becomes linearly separable. Mathematical representation of these kernels is provided in Table 1 and a representation of SVM-based decision boundaries for these kernels is shown in Fig 1.

In this work, several approaches were undertaken to minimize overfitting and improve generalization. For instance, tenfold cross-validation was applied to ensure balanced evaluation across the dataset. The dataset itself was balanced with equal class representation to avoid bias. Additionally, hyperparameters for each model were optimized using grid search to identify the most effective configurations. These methods collectively helped reduce variance and ensure more reliable classification performance.

## Performance metrics

To ensure reliable model evaluation given the limited dataset size, ten-fold cross-validation (CV-10) was used. The classifier's effectiveness was assessed using commonly used evaluation metrics, including accuracy, precision, recall, and F1-score, which are also necessary to comprehensively analyze performance in the presence of class imbalance. These metrics are calculated based on the number of true positive (TP), true negative (TN), false positive (FP), and false negative (FN) predictions, as defined below:

$$Accuracy = \frac{TP + TN}{TP + TN + FP + FN}$$

$$Precision = \frac{TP}{TP + FP}$$

$$Recall = \frac{TP}{TP + FN}$$

$$F - Measure = \frac{2 \times Precision \times Recall}{Precision + Recall}$$

All performance metrics were computed as averages across the folds, providing stable estimates of each model's generalization performance.

**Table 1. SVM kernel functions and their mathematical expressions.**

| Kernel Type | Expression |
|---|---|
| Linear | $x_i^T x_j$ |
| Radial basis function (RBF) | $exp\left(\frac{\|x_i - x_j\|^2}{2\sigma^2}\right)$ |
| Polynomial | $(x_i^T x_j + c)^p$ |
| Sigmoid | $tanh\left(x_i^T x_j + c\right)$ |

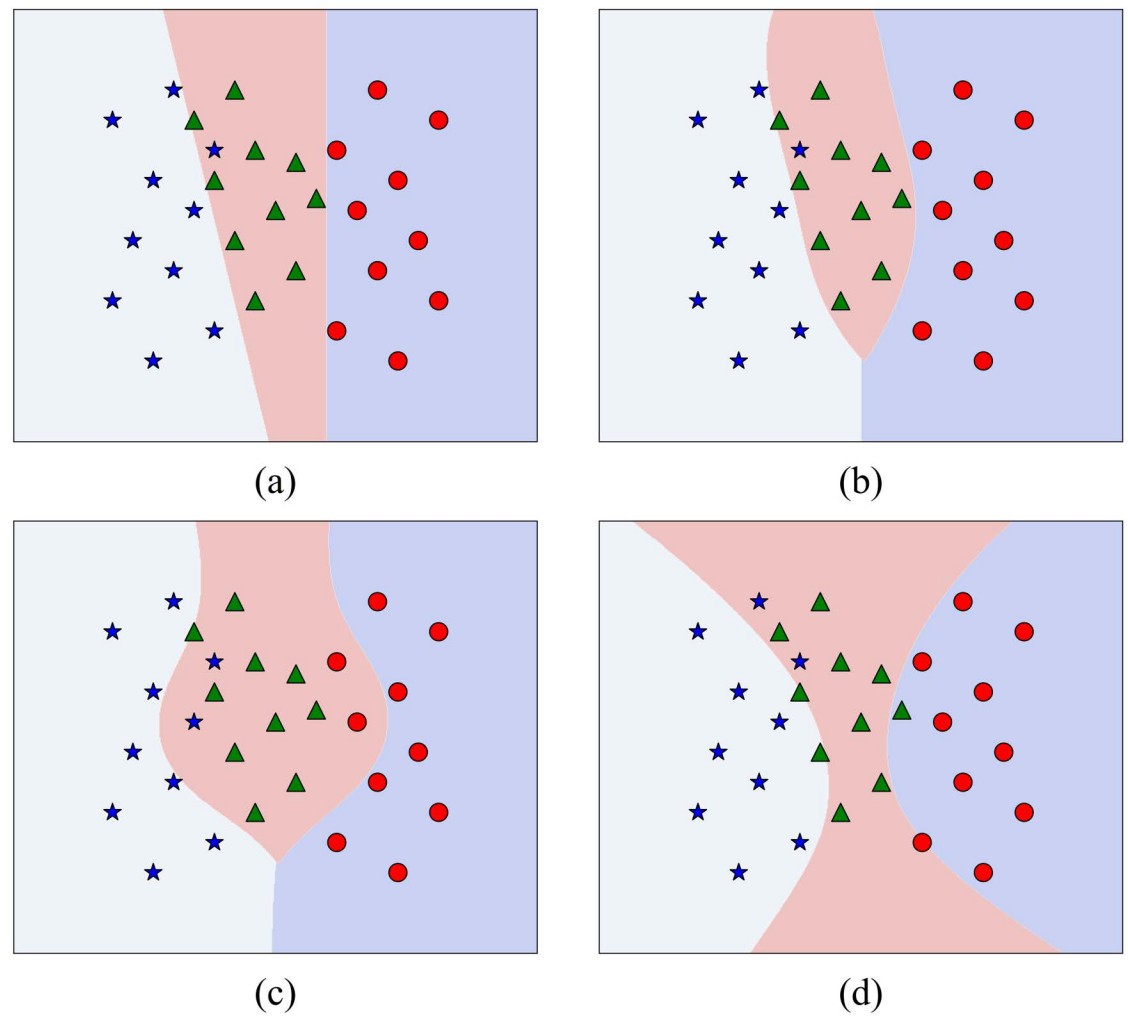

**Fig 1. Decision boundaries of SVM classifiers with different kernel functions: (a) linear, (b) RBF, (c) polynomial, and (d) sigmoid.**

### External validation cohort

To assess the robustness and clinical generalizability of the proposed SVM model, an external validation was conducted using an independent dataset of n = 42 adolescent cricketers collected between 4th to 31st January 2025 from Cricketers Club, Noakhali, This cohort was distinct from the training population. The demographics included 37 males (88.1%) and 5 females (11.9%), with ages ranging from 13 to 19 years (mean age skewed toward 17–19 years). Pain severity was classified using the same criteria as the primary study: No pain (n = 11), Mild (n = 17), and Moderate (n = 14). The pre-trained SVM classifier (Sigmoid kernel, C = 1, gamma = 0.01) was applied to this unseen data without any re-training or parameter tuning.

### Experimental setup

All experiments were conducted using Python 3.13.9 with scikit-learn 1.8.0. Initially, categorical features were converted into numeric labels using LabelEncoder, and subsequently, all features were standardized with StandardScaler to ensure

zero mean and unit variance, allowing fair treatment of all features by the models. Consequently, hyperparameters for each model were optimized using exhaustive grid search combined with stratified k-fold cross-validation, targeting the weighted F1-score to balance performance across all classes. Importantly, during each fold, hyperparameter combinations were evaluated exclusively on the training subset, while the validation subset was kept completely unseen, thereby preventing any data leakage. Once the optimal hyperparameters were identified, the final SVM model was retrained on the full training dataset for evaluation. In parallel, stratified 10-fold cross-validation was employed to preserve the class distribution in each fold, which helped reduce the risk of biased performance estimates. Four ML algorithms, including KNN, RF, LR, and SVM, were evaluated using their best-found configurations. Among them, the SVM classifier, configured with a sigmoid kernel, C = 1, and gamma = 0.01 without class weighting, demonstrated the highest overall performance on the balanced dataset and was therefore selected for detailed analysis.

## Result

The research result was analyzed by two different types of analytical method; among them one was ML-based classification models which included KNN, RF, LR, and SVM model. Another analytical model was binary logistic regression model.

Among the classifiers, SVM achieved the highest overall performance. After extensive grid search-based hyperparameter tuning, the best configuration was found with a sigmoid kernel, C = 1, and gamma = 0.01. As shown in Table 2, SVM attained a test accuracy of 87.6% and a macro-averaged F1 score of 87.1%, outperforming the other models. RF and LR followed closely, with F1 scores of 85.4% and 85.9%, and test accuracies of 85.6% and 86.0%, respectively. KNN, although achieving 100% training accuracy, had the lowest generalization ability, with a test accuracy of 80.4% and an F1 score of 78.9%, indicating overfitting.

To further asses the performance of the SVM classifier in detail, class-wise results were evaluated and are summarized in Table 3. Class 0 (No pain) achieved perfect classification, with 100% precision, recall, and F1-score. Class 1 (Mild pain) was also well classified, with an F1 score of 84.0%. In contrast, Class 2 (Moderate pain) exhibited more classification difficulty, with a precision of 95.5% but a lower recall of 66.0%, resulting in an F1 score of 77.4%. These results indicate that while the SVM model performs well overall, it faces significant challenges in accurately identifying moderate pain cases (Class 2). Clinically, the low recall for Class 2 implies potential under-detection of moderate LBP, which could delay targeted interventions, increasing risks for progression to severe pain or chronicity; this underscores the need for enhanced features (e.g., biomechanical data) in future models to improve triage accuracy.

**Table 2. Overall Algorithm Performance Comparison.**

| Algorithm | Train Acc (%) | Test Acc (%) | Precision (%) | Recall (%) | F1 Score (%) |
|-----------|---------------|--------------|---------------|------------|--------------|
| KNN | **100.0** | 80.4 | 83.7 | 80.4 | 78.9 |
| RF | 96.7 | 85.6 | 86.3 | 85.6 | 85.4 |
| LR | 89.8 | 86.0 | 86.6 | 86.0 | 85.9 |
| **SVM** | 88.4 | **87.6** | **90.0** | **87.6** | **87.1** |

**Table 3. Class-wise Performance Metrics.**

| Class | Train Acc (%) | Test Acc (%) | Precision (%) | Recall (%) | F1 Score (%) |
|-------|---------------|--------------|---------------|------------|--------------|
| 0 | 100.0 | 100.0 | 100.0 | 100.0 | 100.0 |
| 1 | 97.4 | 96.7 | 74.5 | 96.7 | 84.0 |
| 2 | 67.9 | 66.0 | 95.5 | 66.0 | 77.4 |

Visual performance metrics are shown in Fig 2. In Fig 2(a), training and test accuracies across all folds remain closely aligned, with training accuracy around 88–89% and test accuracy between 81–88%, confirming strong generalization and minimal overfitting. Fig 2(b) displays class-wise accuracy across folds. Class 0 achieved perfect accuracy in all folds, while Class 1 remained stable. Class 2 varied more significantly, with fold-wise accuracy ranging from 45% to 80%, underscoring its classification complexity.

Fig 2(c) presents the macro-averaged ROC curve, where SVM achieved an AUC of 0.952, reflecting excellent discrimination among all three classes. The normalized confusion matrix in Fig 2(d) shows 100% correct classification for Class 0, 97% for Class 1, and 66% for Class 2. Misclassification of 34% of Class 2 samples as mild pain illustrates the overlap in features between moderate and mild pain cases and signals a need for enhanced feature differentiation.

**External validation performance**

The SVM model demonstrated strong generalizability on the external cohort (n = 42), achieving an overall accuracy of 81.0% and a weighted F1-score of 0.81. The performance on the unseen data is summarized in Table 4. Consistent with the internal validation, the model achieved perfect classification for Class 0 (No Pain) with 100% precision and recall. For symptomatic cases, the model successfully identified 79% of Moderate Pain (Class 2) cases, which is an improvement over the 66% recall observed during internal cross-validation. However, some overlap persisted, with 21% of moderate cases (3/14) misclassified as mild, and 29% of mild cases (5/17) misclassified as moderate.

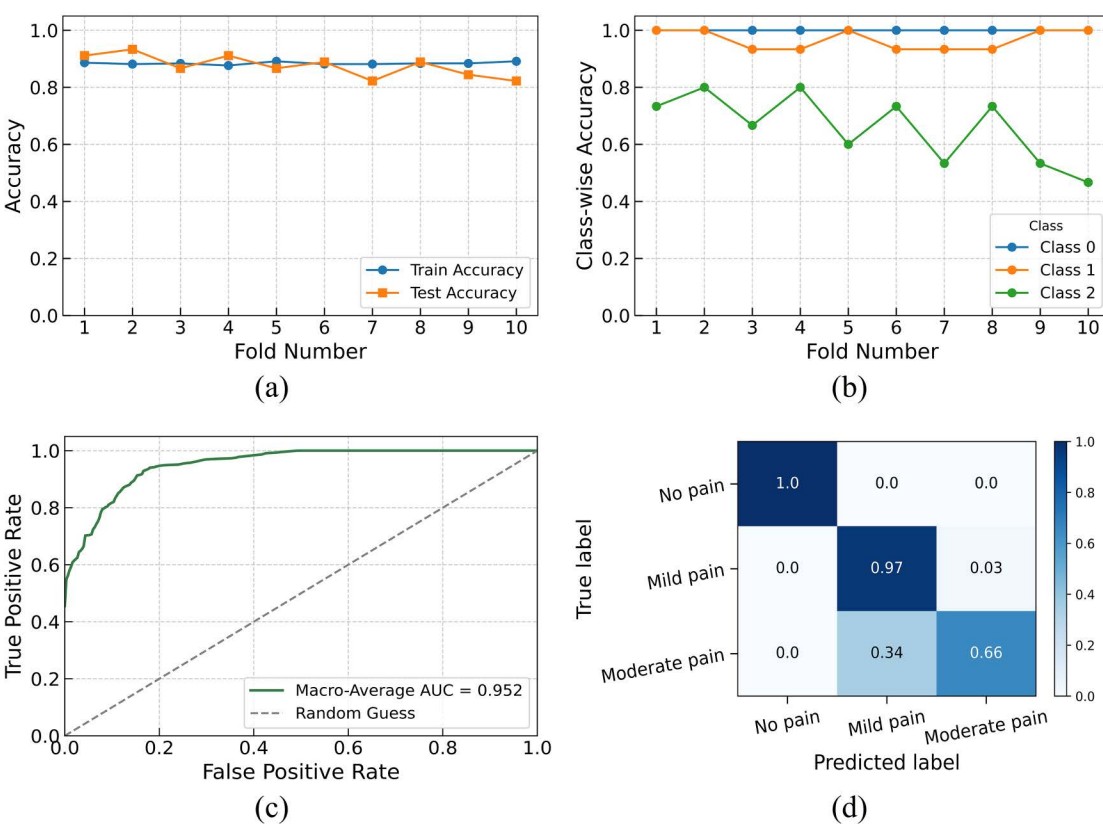

**Fig 2. Performance of the SVM classifier with sigmoid kernel (C = 1, gamma = 0.01) for LBP classification: (a) train and test accuracy across folds, (b) class-wise test accuracy, (c) macro-average ROC curve (AUC = 0.952), and (d) normalized confusion matrix.**

**Table 4. Comparative performance of the SVM model during Internal Cross-Validation vs. External Validation.**

| Performance Metric | | Internal Validation (CV-10) | External Validation (n = 42) | Δ (Change) |
|---|---|---|---|---|
| Overall | Accuracy (%) | 87.6 | 81.0 | -6.6 |
| | Macro-Averaged F1-score (%) | 87.1 | 83.0 | -4.1 |
| Class 0 (No Pain) | Precision (%) | 100.0 | 100.0 | 0.0 |
| | Recall (%) | 100.0 | 100.0 | 0.0 |
| | F1-score (%) | 100.0 | 100.0 | 0.0 |
| Class 1 (Mild Pain) | Precision (%) | 74.5 | 80.0 | +5.5 |
| | Recall (%) | 96.7 | 71.0 | -25.7 |
| | F1-score (%) | 84.0 | 75.0 | -9.0 |
| Class 2 (Moderate Pain) | Precision (%) | 95.5 | 69.0 | -26.5 |
| | Recall (%) | **66.0** | **79.0** | **+13.0** |
| | F1-score (%) | 77.4 | 73.0 | -4.4 |

Furthermore, the performance of the SVM model on the independent external validation dataset (n = 42) is summarized in Table 5. To account for uncertainty due to the relatively small sample size, 95% confidence intervals (CIs) were estimated using bootstrap resampling with 10,000 iterations, where the lower and upper bounds of each interval represent the 2.5th and 97.5th percentiles of the bootstrap distribution, respectively. Overall, the model achieved a weighted F1-score of 81.0% with 95% CI [68.8%, 92.8%], demonstrating reliable performance on unseen data. Class 0 (No Pain) was classified perfectly 100% across all metrics, representing the highest performance, whereas Class 2 (Moderate Pain) exhibited the lowest precision at 68.8% with 95% CI [45.0%, 91.7%], reflecting some overlap in features between mild and moderate pain. The normalized confusion matrix in Fig 3 further illustrates these patterns, with Class 0 (No Pain) correctly classified 100%, Class 1 (Mild Pain) correctly predicted 71%, and Class 2 (Moderate Pain) correctly classified 79%, confirming that the model performs best for the no-pain group and shows some misclassification between moderate and mild pain levels. These results indicate that the SVM model can distinguish between pain levels with quantified uncertainty, even in a small external cohort.

## Sociodemographic, games and preventive measures related characteristics of the participants

This study revealed that several sociodemographic characteristics are significantly associated with the prevalence of low back pain (LBP). A strong association exists for both age (p < 0.001) and educational level (p < 0.001), showing that LBP becomes more common as players get older and advance in their schooling. Monthly family income is also a significant factor (p < 0.001), where participants from the lowest income group reported substantially less LBP than those from higher-income families. Body Mass Index (BMI) showed a significant association (p = 0.009) (Table 6).

Table 7 revealed a highly significant association between LBP and both playing experience (p < 0.001) and the duration of practice per week (p < 0.001). A clear dose-response relationship is visible for both factors. LBP prevalence rises steadily with more years of playing, from 56.4% for those with 1–3 years of experience to 82.5% for those with 7–10

**Table 5. Performance of the SVM model on the external validation dataset with 95% bootstrap confidence intervals.**

| Performance Metric | Precision (%) | Recall (%) | F1-score (%) |
|---|---|---|---|
| Overall (Weighted) | 81.5 [69.7, 93.0] | 81.0 [69.0, 92.9] | 81.0 [68.8, 92.8] |
| Class 0 (No Pain) | 100.0 [100.0, 100.0] | 100.0 [100.0, 100.0] | 100.0 [100.0, 100.0] |
| Class 1 (Mild Pain) | 80.0 [57.1, 100.0] | 70.6 [47.1, 92.3] | 75.0 [54.5, 89.7] |
| Class 2 (Moderate Pain) | 68.8 [45.0, 91.7] | 78.6 [54.5, 100.0] | 73.3 [52.2, 88.9] |

PLOS Digital Health

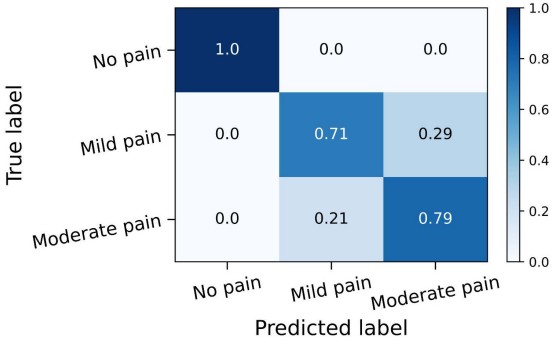

**Fig 3. Normalize confusion matrix of the SVM model on the external validation dataset.**

**Table 6. Distribution of sociodemographic characteristics of the participants (n = 450).**

| Characteristics | Low back pain | | Chi-square | P-value |
|---|---|---|---|---|
| | Present (%) (*n* = 300) | Absent (%) (*n* = 150) | | |
| **Age** | | | | |
| 11-13 years | 25 (44.6) | 31 (55.4) | 19.05 | 0.000[a] |
| 14-16 years | 148 (65.2) | 79 (34.8) | | |
| 17-19 years | 127 (76) | 40 (24) | | |
| **Gender** | | | | |
| Male | 195 (64.3) | 110 (35.7) | 2.49 | 0.132[b] |
| Female | 102 (71.8) | 40 (28.2) | | |
| **Education** | | | | |
| J.S.C/8th grade | 27 (42.2) | 37 (57.8) | 50.03 | 0.000[a] |
| S.S.C/O-levels | 77 (55) | 63 (45) | | |
| H.S.C/A-levels | 83(72.2) | 32 (27.8) | | |
| Under graduation | 113 (86.3) | 18 (13.7) | | |
| **Monthly family income** (BDT)[c] | | | | |
| Below 20000 BDT | 31 (36.9) | 53 (63.1) | 42.82 | 0.000[a] |
| 20000-40000 BDT | 131 (70.4) | 55 (29.6) | | |
| 41000-60000 BDT | 95 (77.2) | 28 (22.8) | | |
| >60000 BDT | 43 (75.4) | 14 (24.6) | | |
| **Body Mass Index** | | | | |
| Underweight | 67 (58.1) | 49 (41.9) | 9.39 | 0.009[a] |
| Normal | 218 (71.2) | 88 (28.8) | | |
| Overweight | 14 (51.9) | 13 (48.1) | | |

[a]Pearson Chi-square; [b]Fisher's exact correction test used for having 20% or more expected frequencies less than 5; [c]1 US dollar = 121.85 BDT (in May 2025)

years. Similarly, LBP prevalence increases with more intense weekly practice, from 47.1% for those training less than 10 hours to 81.7% for those training more than 20 hours.

Table 8 shows of both the frequency of warming up (p < 0.001) and its duration (p < 0.001) were significantly associated with LBP. The players who always warm up had a much lower prevalence of LBP (58.7%) than those who did so

**Table 7. Distribution of games related characteristics of the participants (n = 450).**

| Characteristics | Low back pain | | Chi-square | P-value |
|---|---|---|---|---|
| | Present (%) (n = 300) | Absent (%) (n = 150) | | |
| **Playing position** | | | | |
| Right hand batsman | 118 (66.7) | 59 (33.3) | 4.42 | 0.490[a] |
| Left hand batsman | 24 (60) | 16 (40) | | |
| Spin hand bowler | 23 (56.1) | 18 (43.9) | | |
| Pace hand bowler | 22 (73.3) | 8 (26.7) | | |
| Wicket keeper | 20 (74.1) | 7 (25.9) | | |
| All rounder | 93 (68.9) | 42 (31.1) | | |
| **Playing experience** | | | | |
| 1-3 years | 119 (56.4) | 92 (43.6) | 20.37 | 0.000[a] |
| 4-6 years | 134 (73.6) | 48 (26.4) | | |
| 7-10 years | 47 (82.5) | 10 (17.5) | | |
| **Duration of practice per week** | | | | |
| <10 hours | 24 (47.1) | 27 (52.9) | 27.47 | 0.000[a] |
| 10-20 hours | 151 (61.4) | 95 (38.6) | | |
| >20 hours | 125 (81.7) | 28 (18.3) | | |

[a]Pearson Chi-square; [b]Fisher's exact correction test used for having 20% or more expected frequencies less than 5

**Table 8. Distribution of preventive measure related characteristics of the participants (n = 450).**

| Characteristics | Low back pain | | Chi-square | P-value |
|---|---|---|---|---|
| | Present (%) (n = 300) | Absent (%) (n = 150) | | |
| **Warm-up before sports activity** | | | | |
| Always | 209 (58.7) | 147 (41.3) | 48.58 | 0.000[a] |
| Often | 26 (96.3) | 1 (3.7) | | |
| Sometimes | 65 (97) | 2 (3) | | |
| **Duration of warm-up** | | | | |
| <10 minutes | 77 (52.7) | 69 (47.3) | 19.08 | 0.000[a] |
| 10-15 minutes | 210 (73.7) | 75 (26.3) | | |
| >15 minutes | 13 (68.4) | 6 (31.6) | | |
| **Cool-down after sports activity** | | | | |
| Always | 169 (56.1) | 132 (43.9) | 45.41 | 0.000[a] |
| Often | 16 (84.2) | 3 (15.8) | | |
| Sometimes | 115 (88.5) | 15 (11.5) | | |
| **Duration of cool-down** | | | | |
| <10 minutes | 32 (88.9) | 4 (11.1) | 23.38 | 0.000[a] |
| 10-15 minutes | 219 (69.7) | 95 (30.3) | | |
| >15 minutes | 49 (49.0) | 51 (51.0) | | |

[a]Pearson Chi-square; [b]Fisher's exact correction test used for having 20% or more expected frequencies less than 5

"Often" or "Sometimes" (96.3-97%). For duration, the group warming up for 10–15 minutes reported the highest rate of LBP (73.7%). Likewise, the frequency of cooling down after activity was highly significant (p<0.001), with those who "Always" cool down showing a markedly lower LBP prevalence (56.1%) compared to less consistent players. There is also a significant association between the duration of cool-down and low back pain (LBP) (p<0.001). The lowest prevalence of LBP (30.3%) is observed in the group of players who cool down for 10–15 minutes. In contrast, the prevalence of LBP is considerably higher for those with shorter cool-down periods of less than 10 minutes (61.4%) and is also elevated for those with longer cool-downs of more than 15 minutes (51.0%).

### Logistic regression to predict risk factors of LBP

Table 9 presented the logistic regression analysis and this table revealed several key risk factors for low back pain (LBP) among adolescent cricketers. In the regression model, it was revealed that inconsistent preventive habits were the most powerful predictor. Specifically, cricketers who sometimes (OR = 14.07, 95% CI: 2.78-71.13, p=0.001) or often (OR = 12.48, 95% CI: 1.29-120.22, p=0.029) warmed up had substantially higher odds of experiencing LBP compared to those who always did. Furthermore, the participants who cooled down sometimes nearly tripled the odds of LBP (AOR=2.86, 95% CI: 1.34-6.09, p=0.006). A previous history of LBP also emerged as a major risk factor, increasing the odds of a current episode nearly sevenfold (AOR=6.92, 95% CI: 3.98-12.02, p=0.010). Other significant independent predictors included female gender (AOR=2.52, 95% CI: 1.25-5.07, p=0.010), higher educational attainment as the undergraduate level (AOR=5.38, 95% CI: 1.83-15.76, p=0.002), and higher family income brackets.

To assess the model's fit, the Hosmer-Lemeshow goodness-of-fit test demonstrated that the logistic regression model is effective for forecasting the dependent variable. To evaluate any possible instability due to correlated predictors, variance inflation factors (VIF) were calculated for the regression variables, revealing no notable multicollinearity (VIF<5).

### Discussion

This study utilized a combination of machine learning (ML) classification models and binary logistic regression to analyze risk factors associated with low back pain (LBP) in adolescent cricketers in Dhaka, Bangladesh. The SVM model with a sigmoid kernel demonstrated superior performance, achieving an overall test accuracy of 87.6% and a macro-averaged F1-score of 87.1%, outperforming KNN, RF, and LR. Key risk factors identified through regression included inconsistent warm-up and cool-down practices, prior LBP history, female gender, higher educational attainment, and elevated family income, aligning with multifactorial LBP etiology.

The SVM's strong performance for Class 0 (no pain) and Class 1 (mild pain) underscores its utility in distinguishing lower-severity cases, consistent with prior ML applications in LBP classification, where accuracies range from 83 to 92% in general populations [27]. However, Class 2 (moderate pain) showed lower recall (66%) and F1-score (77.4%), indicating challenges in capturing moderate cases due to feature overlap with mild pain, as evidenced by the confusion matrix. Clinically, this low recall implies a risk of under-detection (false negatives), potentially delaying interventions like targeted physiotherapy or training adjustments, which could lead to progression to severe LBP or chronicity critical in adolescents where early management prevents long-term disability. The external validation confirmed the model's clinical utility, retaining an accuracy of 81.0% on unseen data. This slight decrease from the internal accuracy (87.6%) is expected in machine learning when moving from controlled training sets to real-world data and represents a robust stability. Importantly, the external validation addressed a key concern regarding the detection of moderate pain. While internal testing showed a recall of 66% for Class 2, the external validation achieved a recall of 79%, suggesting the model is highly sensitive to clinically significant pain levels in diverse samples. The confusion matrix of the external data confirms that errors were strictly between adjacent classes (Mild vs. Moderate); no symptomatic players were misclassified as pain-free (Class 0), ensuring that no injured player would be cleared to play erroneously.

**Table 9. Logistic regression for sociodemographic and play related risk factors of low back pain among adolescent cricketers (n = 450).**

| Variables | Category | Low back pain | | | |
|---|---|---|---|---|---|
| | | Absent vs Present | | | |
| | | Unadjusted model | | Adjusted model | |
| | | OR 95% CI | p-value | OR 95% CI | p-value |
| Age | 17-19 years | Reference | | | |
| | 14-16 years | 0.59 (0.37-0.92) | 0.021 | 1.12 (0.60-2.09) | 0.710 |
| | 11-13 years | 0.25 (0.13-0.47) | **<0.001** | 1.52 (0.55-4.21) | 0.418 |
| Gender | Male | Reference | | | |
| | Female | 1.41 (0.91-2.18) | 0.115 | 2.52 (1.25-5.07) | **0.010** |
| Education | J.S.C | Reference | | | |
| | S.S.C/O-levels | 1.67 (0.92-3.04) | 0.091 | 1.85 (0.79-4.33) | 0.154 |
| | H.S.C/A-levels | 3.55 (1.87-6.75) | **<0.001** | 3.75 (1.47-9.57) | **0.006** |
| | Under graduation | 8.60 (4.26-17.36) | **<0.001** | 5.38 (1.83-15.76) | **0.002** |
| Monthly family income (BDT) | <20000 BDT | Reference | | | |
| | 20000-40000 BDT | 4.07 (2.36-7.01) | **<0.001** | 3.12 (1.47-6.63) | **0.003** |
| | 41000-60000 BDT | 5.80 (3.14-10.69) | **<0.001** | 4.69 (1.99-11.04) | **<0.001** |
| | >60000 BDT | 5.25 (2.48-11.09) | **<0.001** | 4.36 (1.58-11.99) | **0.004** |
| Practice duration per week (hour) | <10 hours | Reference | | | |
| | 10-20 hours | 1.78 (0.97-3.28) | 0.060 | 1.21 (0.55-2.63) | 0.623 |
| | >20 hours | 5.02 (2.52-9.97) | **0.000** | 1.70 (0.68-4.22) | 0.249 |
| Warm-up before the sports activity | Always | Reference | | | |
| | Often | 18.28 (2.45-136.26) | **0.005** | 12.48 (1.29-120.22) | **0.029** |
| | Sometimes | 22.85 (5.50-94.83) | **<0.001** | 14.07 (2.78-71.13) | **0.001** |
| Warm-up duration | >15 minutes | Reference | | | |
| | 10-15 minutes | 2.50 (1.65-3.81) | **0.000** | 0.93 (0.53-1.65) | 0.825 |
| | <10 minutes | 1.94 (0.69-5.38) | 0.203 | 0.84 (0.22-3.13) | 0.800 |
| Cool-down after the sports activity | Always | Reference | | | |
| | Often | 4.1 (1.18-14.59) | **0.026** | 2.21 (0.47-10.31) | 0.313 |
| | Sometimes | 5.98 (3.33-10.74) | **<0.001** | 2.86 (1.34-6.09) | **0.006** |
| Cool-down duration | >15 minutes | Reference | | | |
| | 10-15 minutes | 2.39 (1.51-3.80) | **<0.001** | 1.63 (0.88-2.98) | 0.114 |
| | <10 minutes | 8.32 (2.74-25.28) | **<0.001** | 2.65 (0.65-10.78) | 0.173 |
| Pervious history of LBP | Absent | Reference | | | |
| | Present | 7.52 (4.85-11.67) | **<0.001** | 6.92 (3.98-12.02) | **0.010** |

Findings on sociodemographic risks, such as Female gender was found to be an independent risk factor (AOR = 2.52, p = 0.010), aligning with general epidemiological trends where females often report higher rates of LBP, potentially due to anatomical, hormonal, or biomechanical differences. A large French occupational health study found female gender associated with increased risk of musculoskeletal disorder-related work unfitness (OR = 2.09) [28].

This research revealed a robust and statistically significant correlation between education level and low back pain (LBP) (P < 0.001). Individuals with undergraduate degrees displayed a higher LBP prevalence of 86.3% relative to those with less education. Likewise, a Chinese investigation involving 15,743 people identified 1.24 times greater odds of LBP among those with higher education levels [29].

In this study it was found that players who practiced > 20 hours/week had the highest prevalence of LBP (81.7%), while those who practiced <10 hours/week had the lowest prevalence (47.1%). Similarly, in a research, adolescent basketball players practicing over 6.6 hours/week exhibited 43.8% LBP prevalence [30] and youth weightlifters training long-term weightlifting developed lumbar disc degeneration within 4 years, progressing to herniation in 33% by year 5 [31].

A strong association was found between warming up before sports activity and LBP (P < 0.001), with those who always warm up having a lower prevalence of LBP (56.6%). Similarly, a meta-analysis of Warm-up Intervention Programs (WIPs) found a 36% reduction in sports injuries when accounting for hours of risk exposure [32].

Similarly, strong association was also observed between cool-down after sports activity and LBP (P < 0.001), with those who always perform a cool-down having a lower prevalence of LBP (56.1%). Similarly, a randomized crossover study compared aqua- and land-based cool-down exercises found both have similar recovery effects on muscle soreness and performance-based parameters [33].

Previous history of LBP was a consistent and strong predictor of current LBP, with participants who had a prior history of LBP showing higher odds of presenting LBP (OR = 7.52). A comprehensive systematic review investigating the risk of LBP recurrence found that "a history of previous episodes of LBP prior to the most recent episode was the only factor that consistently predicted recurrence of LBP" [34]. Another systematic review also discovered intrinsic factors related to LBP in cricket fast bowlers to include previous injury as a factor known to influence load tolerance and LBP injury [17].

The application of machine learning here is a useful sports medicine innovation. Methods of traditional risk factor analysis cannot account for such complex, non-linear interdependencies amongst a large number of variables such as sociodemographic, game and prevention-of-injury factors and LBP factors.

## Conclusion

The aim of this study was to perform a risk assessment and classification of LBP among adolescent cricketers in Dhaka City, Bangladesh. This was an effort to fill an existing knowledge gap through ML techniques to derive key risk factors of LBP in this population. A wide range of variables were considered, including sociodemographic, game-related, preventive measures, and LBP-related variables, to evaluate model performance. A regression model was proposed to predict LBP risk factors, while a SVM model was identified as the most suitable ML classifier for the classification of LBP levels. LBP was classified into three levels, i.e., no pain, mild pain, and moderate pain, excluding severe pain category due to a limited number of severe cases. Despite the overall satisfactory performance, results show that identifying Class 2 (moderate pain) remains challenging. This may be due to overlapping clinical features between mild and moderate pain that limit separability. The findings provide significant insight to health professionals, coaches, and sport organizations in designing effective interventions to prevent and treat LBP in young players.

### Limitation

This study has several limitations. The exclusion of a severe pain category due to insufficient data reduces the model's depth and rigor, limiting its capacity to triage urgent cases involving high biomechanical stress or chronic progression. Purposive sampling from specific Dhaka clubs introduces selection bias, potentially restricting generalizability to broader populations like rural or non-club-based players, while reliance on self-reported data may cause recall or social desirability bias, especially among adolescents normalizing pain amid cultural influences. While external validation was performed, the validation cohort was smaller than the training set and predominantly male, which may limit inferences regarding female athletes in the validation phase.

## Supporting information

**S1 Appendix. Informed consent form.** The written consent form was administered to the participants.
(DOCX)

**S2 Appendix. Questionnaire.** The included questions comprised the questionnaire administered to the participants.
(DOCX)

**S3 Appendix. Data.** This file contains primary research data.
(CSV)

**S4 Appendix. Data.** This file contains research data for external validation.
(CSV)

## Acknowledgments

The authors would like to thank coaches and players of BKSP, Lt. Sheikh Jamal cricket Academy, Kola Bagan Krira chakra, Khelaghar Cricket Academy, and City club for all assistance and cooperation throughout the study.

## Author contributions

**Conceptualization:** Marzana Afrooj Ria, Tasrima Trisha Ratna, Shudeshna Chakraborttye Purba, Rubal Kar, Mohoshina Karim, Md Osman Ali, Erfat Jaren Chaity, Shahadath Hossen, Joynal Abedin Imran, Shahriar Hasan.

**Data curation:** Erfat Jaren Chaity, Shahriar Hasan.

**Formal analysis:** Tasrima Trisha Ratna, Rubal Kar.

**Investigation:** Marzana Afrooj Ria, Shahriar Hasan.

**Methodology:** Marzana Afrooj Ria, Shudeshna Chakraborttye Purba, Md Osman Ali.

**Project administration:** Mohoshina Karim, Joynal Abedin Imran.

**Resources:** Rubal Kar, Mohoshina Karim, Erfat Jaren Chaity.

**Software:** Md Osman Ali, Shahadath Hossen, Joynal Abedin Imran.

**Supervision:** Marzana Afrooj Ria, Shudeshna Chakraborttye Purba, Shahriar Hasan.

**Validation:** Tasrima Trisha Ratna, Shudeshna Chakraborttye Purba.

**Visualization:** Rubal Kar, Md Osman Ali, Shahadath Hossen.

**Writing – original draft:** Marzana Afrooj Ria.

**Writing – review & editing:** Mohoshina Karim, Md Osman Ali, Shahadath Hossen.

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
