## [Decision Letter · Decision Letter 0]

16 Sep 2025

Response to Reviewers'. This file does not need to include responses to any formatting updates and technical items listed in the 'Journal Requirements' section below.'. This file does not need to include responses to any formatting updates and technical items listed in the 'Journal Requirements' section below.* A marked-up copy of your manuscript that highlights changes made to the original version. You should upload this as a separate file labeled 'Revised Manuscript with Track Changes'.'.* An unmarked version of your revised paper without tracked changes. You should upload this as a separate file labeled 'Manuscript'.'. If you would like to make changes to your financial disclosure, competing interests statement, or data availability statement, please make these updates within the submission form at the time of resubmission. Guidelines for resubmitting your figure files are available below the reviewer comments at the end of this letter. We look forward to receiving your revised manuscript. Kind regards, Cleva Villanueva, M.D., Ph.D.Academic EditorPLOS Digital Health Cleva VillanuevaAcademic EditorPLOS Digital Health Leo Anthony CeliEditor-in-ChiefPLOS Digital Healthorcid.org/0000-0001-6712-6626 **Journal Requirements:**

1. Please provide an Author Summary. This should appear in your manuscript between the Abstract (if applicable) and the Introduction, and should be 150–200 words long. The aim should be to make your findings accessible to a wide audience that includes both scientists and non-scientists. Sample summaries can be found on our website under Submission Guidelines:

https://journals.plos.org/digitalhealth/s/submission-guidelines#loc-parts-of-a-submission

2. We note that there is identifying data in the Supporting Information file < Data.csv>. Due to the inclusion of these potentially identifying data, we have removed this file from your file inventory. Prior to sharing human research participant data, authors should consult with an ethics committee to ensure data are shared in accordance with participant consent and all applicable local laws.

-Location data

**Additional Editor Comments (if provided):** Reviewer #1: Abstract: Needs clearer statement of feature correlates and explicit mention of study limitations. Absence of the severe pain category reduces the model’s capacity to triage urgent cases.

Introduction: Should include local statistics on LBP in Bangladesh cricketers, acknowledge genetic predisposition as a factor, and cite comparative studies (traditional vs. ML approaches). Needs references to existing ML work on LBP and discussion of ML’s achievements in other contexts.

Methods: Adequate sample size (n=450) and sound justification for regression models. However, purposive sampling may cause bias, reliance on self-reported data is limiting, no feature selection analysis mentioned, unclear categorical variable encoding, and absence of external validation.

Results: Well presented, but tables could be shorter. Authors should discuss the clinical implications of low recall in Class 2 (66%).

Discussion: Strong comparative analysis, but requires deeper discussion on clinical significance of Class 2 low recall and external validation for generalizability.

Overall: A relevant study with strong potential, but would be significantly strengthened by external validation, refined methodology, and clearer justification in introduction.

Reviewer #2: Introduction: Too long and should be condensed to fewer than 400 words without omitting important content.

Methods: Missing inclusion/exclusion criteria (e.g., training duration per week required for eligibility). Avoid reporting model performance in the Methods section—this belongs in Results.

Results: First paragraph should instead describe participant demographics. Authors should follow a standardized checklist (e.g., STROBE) for reporting cross-sectional studies.

Formatting: Inconsistent numerical reporting (percentages in text, decimals in tables) should be standardized.

Overall: Main issues relate to manuscript organization, clarity, and adherence to reporting standards.

Reviewer #3: Overall Premise: Interesting use of ML for a low-cost solution, but key conceptual issues remain.

Main Concerns:

Lack of clarity on how athletes were classified into “no, mild, and moderate LBP” despite existing literature.

No discussion on potential collinearity of variables, which may compromise model stability.

Risk factors identified are already well-established in prior studies—unclear what the novel contribution is.

Practical application of the algorithm in real-world settings is not addressed.

Introduction: Some references inappropriate (e.g., citation of qualitative/lived experience study as demographic evidence). Authors should use more robust sources (large cohort studies, meta-analyses, RCTs). Questionable logical connections in cited arguments.

Methods: Lack of clarity on who categorized LBP severity, which standard was used, and what reference supports it.

**Reviewers' Comments:** Reviewer's Responses to Questions

**Comments to the Author**

1. Does this manuscript meet PLOS Digital Health’s publication criteria? Is the manuscript technically sound, and do the data support the conclusions? The manuscript must describe methodologically and ethically rigorous research with conclusions that are appropriately drawn based on the data presented.? Is the manuscript technically sound, and do the data support the conclusions? The manuscript must describe methodologically and ethically rigorous research with conclusions that are appropriately drawn based on the data presented.

Reviewer #1: Partly

Reviewer #2: No

Reviewer #3: Yes

2. Has the statistical analysis been performed appropriately and rigorously?

Reviewer #1: No

Reviewer #2: Yes

Reviewer #3: No

3. Have the authors made all data underlying the findings in their manuscript fully available (please refer to the Data Availability Statement at the start of the manuscript PDF file)?

The PLOS Data policy requires authors to make all data underlying the findings described in their manuscript fully available without restriction, with rare exception. The data should be provided as part of the manuscript or its supporting information, or deposited to a public repository. For example, in addition to summary statistics, the data points behind means, medians and variance measures should be available. If there are restrictions on publicly sharing data—e.g. participant privacy or use of data from a third party—those must be specified.requires authors to make all data underlying the findings described in their manuscript fully available without restriction, with rare exception. The data should be provided as part of the manuscript or its supporting information, or deposited to a public repository. For example, in addition to summary statistics, the data points behind means, medians and variance measures should be available. If there are restrictions on publicly sharing data—e.g. participant privacy or use of data from a third party—those must be specified.

Reviewer #1: Yes

Reviewer #2: No

Reviewer #3: Yes

4. Is the manuscript presented in an intelligible fashion and written in standard English?

Reviewer #1: No

Reviewer #2: Yes

Reviewer #3: Yes

Reviewer #1: Review of Machine Learning-Based Pattern Recognition of Risk Factors for Low Back Pain among Adolescent Cricket Players in Dhaka City

Abstract

A Relevant study with clear focus

The abstract section will be more helpful if the feature correlates were clearly stated.

The study acknowledged the absence of the fourth class (Severe pain category) impacting the depth and rigor of the study.

(The model developed by this study may have reduced capacity to triage LBP and determine cricket players who need urgent intervention especially the severe pain category). The circumstances is understandable albeit.

The limitations of this study should be summarized in the abstract.

Introduction

The authors should add compelling local statistics demonstrating the severity of LBP amongst cricketers in Bangladesh or is there a lack of studies exploring this ? Which then may be a solid justification for the existence of this study.

The study acknowledged the strong possibility of the under-reportage of LBP in Bangladesh

The authors did not acknowledge the fact that genetic predisposition also account for LBP, it isn’t all mechanical.

The study will benefit from citation of comparative studies where traditional methods have shown limited capacity which will strengthen the justification of Machine learning for this study.

”There are no known predictive models with ML to be used in screening LBP in adolescent cricketers in Dhaka “ Other studies on LBP among cricketers using ML for predictive analysis should be cited to justify the need to carry out this study in Bangladesh

Also, the study should highlight statistically how the performance and effectiveness of ML in predicting the occurrence of LBP amongst cricketers in other places, what achievements so far? Has it reduced the occurrence?

Materials and Method:

n = 450 seems adequate for this study

Ethical consideration clearly stated

Purposive sampling may introduce selection bias

The study demonstrated heavy reliance on self reported data and also, no mention of feature selection analysis

How were the categorical variables encoded?

The study could have benefited from an external validation

The method section provided a good justification for the use of the regression models and dropping the deep learning model considering the small sample size.

The inclusion of the regression models for comparative analysis makes this study more robust

The use of a balanced dataset, cross validation and grid search reduces overfitting which makes the study more robust.

Results.

Clear presentation of performance metrics of the models employed for the study

Table can be more abridged

For class 2, a low recall value of 66% and F1 value of 77% was achieved by the SVM classifier. Can the authors address the clinical implication of this?

The study has a visual appeal (Visualization of results)

Discussion

This chapter features a robust analysis of the performance metrics and good comparative analysis of the study with other studies on LBP.

The authors should have a more robust discussion on the clinical significance of the class 2 low recall value and what steps to take to mitigate it.

There is a need for the authors to conduct an external validation of their model to assess generalizability and performance on unseen data.

Overall: A very germane study that would be more impactful with an external validation and more refined methodology.

Reviewer #2: - Introduction is too long and can be summarized without deletion of important points. please re-write the introduction in less than 400 words.

- please report the inclusion and exclusion criteria; for example, what was the minimum training duration per week needed to enter the study.

- please avoid reporting the performance of different models in method section. for example: "Although it offered interpretable baseline results, it was less effective than more robust classifiers like SVM in capturing complex relationships between low back pain and its contributing factors". this information belongs to result section

- The first paragraph of results belongs to reporting the demographics of included subjects. please use a standard check list such as "STROBE Statement—Checklist of items that should be included in reports of cross-sectional studies"

- It is advised to report the numbers in one format. in this manuscript the numbers are reported as percentages in text but as decimals in tables.

Reviewer #3: The authors provided an interesting premise on using machine learning to provide low cost solution for assessing low back pain (LBP) risk among young cricket athletes in Dhaka. Specific cricket fast bowling techniques are well known to increase risk of neural arch fracture due to the biomechancial load on the spine. My main concern is that the authors did not discuss how they even classify these athletes into no, mild and moderate LBP, when multiple reviews have been done regarding this subject. Another main issue is lack of discussion how some of the variables used in this study might be highly correlated, which is known to cause instability on a predictive model. Last but not least, the risk factors identified in this study had previously been discovered in prior study, so what is the added value of this study? How would the algorithm be used in real life setting?

Below are more detailed comments on this paper:

Intro

- Line 5: up to 30% of adolescents affected. The cited reference 2 is not a demographic study but qualitative study regarding young athlete’s lived in experience. not appropriate for this statement

- Ref 4 narative review, ref 7 is case report, ref 8 is scoping review. Can you provide more objective sources of information such as large demographic studies, network meta analysis or randomized controlled trials?

- Line 34: How does argument 6 supports the argument about the scarcity of biomechanical assessment in Dhaka?

Methods

- Line 60-64: Who categorized the LBP severity, based on what standard and what is the related reference used to use this standard?

- Line 75-99 provide justifications for different types of participant classifications such as height, weight, all the way to playing position with no reference on why these are important relative to CLBP.

- Line 126-7: “ it was less effective in capturing nonlinear associations compared to tree-based or kernel-based models.” Any reference for this?

- Overall, please cite proper references as basis of your methodology

Results

- Some of these factors are highly correlated to each other, such as age and level of education within recruited demographics, assuming none of the recruited participants will drop out on elementry. Use of highly correlated variables to create a predictive model can create an unstable model. How come there’s no assessment on the variable correlation?

Discussion

line 292-296: your results may appear to contradict reference 19, when reference 19 refers to physical education that not all participants received, while your education level may be strongly correlated with age. Receiving education is not equal to receiving physical education.

**Do you want your identity to be public for this peer review?** For information about this choice, including consent withdrawal, please see our Privacy Policy..

Reviewer #1: No

Reviewer #2: **Yes:** Hamidreza AshayeriHamidreza AshayeriHamidreza AshayeriHamidreza Ashayeri

Reviewer #3: No

**Figure resubmission:** While revising your submission, we strongly recommend that you use PLOS’s NAAS tool (https://ngplosjournals.pagemajik.ai/artanalysis) to test your figure files. NAAS can convert your figure files to the TIFF file type and meet basic requirements (such as print size, resolution), or provide you with a report on issues that do not meet our requirements and that NAAS cannot fix. 

**Reproducibility:** To enhance the reproducibility of your results, we recommend that authors of applicable studies deposit laboratory protocols in protocols.io, where a protocol can be assigned its own identifier (DOI) such that it can be cited independently in the future. Additionally, PLOS ONE offers an option to publish peer-reviewed clinical study protocols. Read more information on sharing protocols at https://plos.org/protocols?utm_medium=editorial-email&utm_source=authorletters&utm_campaign=protocols To enhance the reproducibility of your results, we recommend that authors of applicable studies deposit laboratory protocols in protocols.io, where a protocol can be assigned its own identifier (DOI) such that it can be cited independently in the future. Additionally, PLOS ONE offers an option to publish peer-reviewed clinical study protocols. Read more information on sharing protocols at https://plos.org/protocols?utm_medium=editorial-email&utm_source=authorletters&utm_campaign=protocols

---

## [Decision Letter · Decision Letter 1]

26 Dec 2025

Response to Reviewers'. This file does not need to include responses to any formatting updates and technical items listed in the 'Journal Requirements' section below.'. This file does not need to include responses to any formatting updates and technical items listed in the 'Journal Requirements' section below.* A marked-up copy of your manuscript that highlights changes made to the original version. You should upload this as a separate file labeled 'Revised Manuscript with Track Changes'.'.* An unmarked version of your revised paper without tracked changes. You should upload this as a separate file labeled 'Manuscript'.'. If you would like to make changes to your financial disclosure, competing interests statement, or data availability statement, please make these updates within the submission form at the time of resubmission. Guidelines for resubmitting your figure files are available below the reviewer comments at the end of this letter. We look forward to receiving your revised manuscript.  Kind regards, Cleva Villanueva, M.D., Ph.D.Academic EditorPLOS Digital Health Cleva VillanuevaAcademic EditorPLOS Digital Health Leo Anthony CeliEditor-in-ChiefPLOS Digital Healthorcid.org/0000-0001-6712-6626  **Journal Requirements:** If the reviewer comments include a recommendation to cite specific previously published works, please review and evaluate these publications to determine whether they are relevant and should be cited. There is no requirement to cite these works unless the editor has indicated otherwise.   **Additional Editor Comments:** We appreciate the effort made to revise the manuscript and to address several of the reviewers’ comments. The revised version shows improvements; however, an important methodological concern remains unresolved.

Specifically, the manuscript lacks external validation of the proposed model. While the authors state that they do not currently have the resources to perform external validation and mention this as future work, external validation is a key requirement to ensure the reproducibility, robustness, and generalizability of the results. Even a limited or temporary external validation (e.g., using an independent dataset, a temporal split, or data from a different setting) would substantially strengthen the manuscript.

Without external validation, the conclusions remain insufficiently supported for publication at this stage. We therefore encourage the authors to address this issue directly by providing an appropriate form of external validation or, alternatively, by clearly justifying why such validation cannot be performed and revising the scope and claims of the manuscript accordingly.

**Reviewers' Comments:**Reviewer's Responses to Questions

**Comments to the Author**

Reviewer #1: (No Response)

Reviewer #2: All comments have been addressed

publication criteria? Is the manuscript technically sound, and do the data support the conclusions? The manuscript must describe methodologically and ethically rigorous research with conclusions that are appropriately drawn based on the data presented.? Is the manuscript technically sound, and do the data support the conclusions? The manuscript must describe methodologically and ethically rigorous research with conclusions that are appropriately drawn based on the data presented.

Reviewer #1: No

Reviewer #2: Partly

3. Has the statistical analysis been performed appropriately and rigorously?

Reviewer #1: No

Reviewer #2: Yes

4. Have the authors made all data underlying the findings in their manuscript fully available (please refer to the Data Availability Statement at the start of the manuscript PDF file)?

The PLOS Data policy requires authors to make all data underlying the findings described in their manuscript fully available without restriction, with rare exception. The data should be provided as part of the manuscript or its supporting information, or deposited to a public repository. For example, in addition to summary statistics, the data points behind means, medians and variance measures should be available. If there are restrictions on publicly sharing data—e.g. participant privacy or use of data from a third party—those must be specified.requires authors to make all data underlying the findings described in their manuscript fully available without restriction, with rare exception. The data should be provided as part of the manuscript or its supporting information, or deposited to a public repository. For example, in addition to summary statistics, the data points behind means, medians and variance measures should be available. If there are restrictions on publicly sharing data—e.g. participant privacy or use of data from a third party—those must be specified.

Reviewer #1: Yes

Reviewer #2: Yes

5. Is the manuscript presented in an intelligible fashion and written in standard English?

Reviewer #1: Yes

Reviewer #2: Yes

Reviewer #1: Thanks for the corrections added but it will be nice if this model was externally validated. This seems like the only missing piece in the puzzle.

Reviewer #2: I`m grateful for the chance to review the revised version of this manuscript. The authors have addressed the reviewers' comments, and I appreciate their dedication.

They have included essential statistical analyses, such as multicollinearity testing, which enhances the value of their work. They have also applied fundamental revisions in the main text, especially in the discussion section, improving the content of their manuscript.

Their work may prove beneficial for the application of artificial intelligence in low-resource areas.

**Do you want your identity to be public for this peer review?** For information about this choice, including consent withdrawal, please see our Privacy Policy..

Reviewer #1: No

Reviewer #2: **Yes:** Hamidreza AshayeriHamidreza AshayeriHamidreza AshayeriHamidreza Ashayeri

**Figure resubmission:** While revising your submission, we strongly recommend that you use PLOS’s NAAS tool (https://ngplosjournals.pagemajik.ai/artanalysis) to test your figure files. NAAS can convert your figure files to the TIFF file type and meet basic requirements (such as print size, resolution), or provide you with a report on issues that do not meet our requirements and that NAAS cannot fix. 

**Reproducibility:** To enhance the reproducibility of your results, we recommend that authors of applicable studies deposit laboratory protocols in protocols.io, where a protocol can be assigned its own identifier (DOI) such that it can be cited independently in the future. Additionally, PLOS ONE offers an option to publish peer-reviewed clinical study protocols. Read more information on sharing protocols at https://plos.org/protocols?utm_medium=editorial-email&utm_source=authorletters&utm_campaign=protocols To enhance the reproducibility of your results, we recommend that authors of applicable studies deposit laboratory protocols in protocols.io, where a protocol can be assigned its own identifier (DOI) such that it can be cited independently in the future. Additionally, PLOS ONE offers an option to publish peer-reviewed clinical study protocols. Read more information on sharing protocols at https://plos.org/protocols?utm_medium=editorial-email&utm_source=authorletters&utm_campaign=protocols

---

## [Decision Letter · Decision Letter 2]

25 Feb 2026

Response to Reviewers'. This file does not need to include responses to any formatting updates and technical items listed in the 'Journal Requirements' section below.'. This file does not need to include responses to any formatting updates and technical items listed in the 'Journal Requirements' section below.* A marked-up copy of your manuscript that highlights changes made to the original version. You should upload this as a separate file labeled 'Revised Manuscript with Track Changes'.'.* An unmarked version of your revised paper without tracked changes. You should upload this as a separate file labeled 'Manuscript'.'. If you would like to make changes to your financial disclosure, competing interests statement, or data availability statement, please make these updates within the submission form at the time of resubmission. Guidelines for resubmitting your figure files are available below the reviewer comments at the end of this letter. We look forward to receiving your revised manuscript. Kind regards, Cleva Villanueva, M.D., Ph.D.Academic EditorPLOS Digital Health Cleva VillanuevaAcademic EditorPLOS Digital Health Leo Anthony CeliEditor-in-ChiefPLOS Digital Healthorcid.org/0000-0001-6712-6626  **Journal Requirements:** If the reviewer comments include a recommendation to cite specific previously published works, please review and evaluate these publications to determine whether they are relevant and should be cited. There is no requirement to cite these works unless the editor has indicated otherwise.  **Additional Editor Comments (if provided):** Dear Authors,

Your manuscript has significantly improved, and the revisions have addressed the major methodological concerns raised in the previous round of review. In particular, the addition of an independent external validation cohort (n = 42) substantially strengthens the study and directly resolves the key gap identified earlier. The revised manuscript is now very close to being publication-ready.

Thank you for the thorough and thoughtful revision.

There are, however, a few remaining points that should be addressed prior to acceptance:

1. Reporting uncertainty in external validation

Please include measures of uncertainty for the performance metrics obtained from the external cohort (e.g., 95% confidence intervals via bootstrap methods or exact binomial CIs for class recalls). Given the relatively small sample size (n = 42), performance estimates may be unstable. Reporting confidence intervals will ensure appropriate calibration of the claims. Additionally, consider including the external confusion matrix in the main text or Supplementary Materials.

2. Clarification of hyperparameter tuning and preprocessing

Please clarify how grid search and preprocessing were conducted relative to cross-validation. If hyperparameters were tuned on the full dataset prior to cross-validation, reported performance may be optimistic. A brief statement confirming that tuning occurred within training folds (e.g., nested cross-validation or equivalent) would resolve this concern.

3. Dataset balancing procedure

The manuscript states that the dataset is balanced (150 samples per class), but the method and stage at which balancing was performed are not fully explained. Please specify whether under-sampling, over-sampling, or class weighting was used. If feasible, consider including a brief sensitivity analysis using the original class distribution (or class weights) to demonstrate robustness.

4. Data Availability statement

There appears to be inconsistent language regarding whether the dataset is fully available within the manuscript/Supplementary Information or “available upon request.” PLOS requires that the underlying data necessary to replicate the results be publicly accessible without restriction, except in rare and justified cases. Please ensure that the final Data Availability statement is fully compliant and consistent throughout the manuscript.

**Reviewers' Comments:** Reviewer's Responses to Questions

**Comments to the Author**

Reviewer #1: All comments have been addressed

Reviewer #4: (No Response)

publication criteria? Is the manuscript technically sound, and do the data support the conclusions? The manuscript must describe methodologically and ethically rigorous research with conclusions that are appropriately drawn based on the data presented.? Is the manuscript technically sound, and do the data support the conclusions? The manuscript must describe methodologically and ethically rigorous research with conclusions that are appropriately drawn based on the data presented.

Reviewer #1: Yes

Reviewer #4: No

3. Has the statistical analysis been performed appropriately and rigorously?

Reviewer #1: Yes

Reviewer #4: Yes

4. Have the authors made all data underlying the findings in their manuscript fully available (please refer to the Data Availability Statement at the start of the manuscript PDF file)?

The PLOS Data policy requires authors to make all data underlying the findings described in their manuscript fully available without restriction, with rare exception. The data should be provided as part of the manuscript or its supporting information, or deposited to a public repository. For example, in addition to summary statistics, the data points behind means, medians and variance measures should be available. If there are restrictions on publicly sharing data—e.g. participant privacy or use of data from a third party—those must be specified.requires authors to make all data underlying the findings described in their manuscript fully available without restriction, with rare exception. The data should be provided as part of the manuscript or its supporting information, or deposited to a public repository. For example, in addition to summary statistics, the data points behind means, medians and variance measures should be available. If there are restrictions on publicly sharing data—e.g. participant privacy or use of data from a third party—those must be specified.

Reviewer #1: Yes

Reviewer #4: No

5. Is the manuscript presented in an intelligible fashion and written in standard English?

Reviewer #1: Yes

Reviewer #4: Yes

Reviewer #1: All concerns have been addressed.

Reviewer #4: Thank you for the thorough revision. The addition of an independent external validation cohort (n = 42) substantially strengthens the work and directly addresses the key methodological gap raised previously. The revised manuscript is much closer to being publication-ready.

Key remaining points to address:

1. Please add uncertainty around performance on the external cohort (for example 95% CIs via bootstrap or exact binomial CIs for class recalls). With n = 42, estimates can be unstable, and reporting CIs will make the claims appropriately calibrated. Consider also showing the external confusion matrix in the main text or supplement.

2. Clarify exactly how grid search and preprocessing were conducted relative to cross-validation. If hyperparameters were tuned on the full dataset before CV, reported CV performance may be optimistic. A brief statement confirming tuning within training folds (nested CV or equivalent) would resolve this.

3. The text states the dataset is balanced (150 per class) but does not fully explain how balancing was achieved and at what step. Please specify whether you used under-sampling, over-sampling, or weighting, and consider a short sensitivity analysis using the original class distribution (or class weights) to show robustness.

4. The manuscript contains potentially conflicting language about data being in the manuscript/SI vs “available on request.” PLOS requires the underlying data needed to replicate results to be publicly available without restriction except rare justified cases, so please ensure the final Data Availability statement is fully compliant and consistent throughout.

**Do you want your identity to be public for this peer review?** For information about this choice, including consent withdrawal, please see our Privacy Policy..

Reviewer #1: No

Reviewer #4: No

**Figure resubmission:**  While revising your submission, we strongly recommend that you use PLOS’s NAAS tool (https://ngplosjournals.pagemajik.ai/artanalysis) to test your figure files. NAAS can convert your figure files to the TIFF file type and meet basic requirements (such as print size, resolution), or provide you with a report on issues that do not meet our requirements and that NAAS cannot fix. 

**Reproducibility:** To enhance the reproducibility of your results, we recommend that authors of applicable studies deposit laboratory protocols in protocols.io, where a protocol can be assigned its own identifier (DOI) such that it can be cited independently in the future. Additionally, PLOS ONE offers an option to publish peer-reviewed clinical study protocols. Read more information on sharing protocols at https://plos.org/protocols?utm_medium=editorial-email&utm_source=authorletters&utm_campaign=protocols To enhance the reproducibility of your results, we recommend that authors of applicable studies deposit laboratory protocols in protocols.io, where a protocol can be assigned its own identifier (DOI) such that it can be cited independently in the future. Additionally, PLOS ONE offers an option to publish peer-reviewed clinical study protocols. Read more information on sharing protocols at https://plos.org/protocols?utm_medium=editorial-email&utm_source=authorletters&utm_campaign=protocols

---

## [Editor Report · Decision Letter 3]

30 Mar 2026

Machine Learning-Based Pattern Recognition of Risk Factors for Low Back Pain among Adolescent Cricket Players in Dhaka City

PDIG-D-25-00567R3

Dear Marzana Afrooj Ria,

We are pleased to inform you that your manuscript 'Machine Learning-Based Pattern Recognition of Risk Factors for Low Back Pain among Adolescent Cricket Players in Dhaka City' has been provisionally accepted for publication in PLOS Digital Health.

Best regards,

Cleva Villanueva, M.D., Ph.D.

Academic Editor

PLOS Digital Health

**Additional Editor Comments (if provided):**

After carefully reviewing the original manuscript, the revised versions, and the reviewers’ comments, the editor has decided to accept the manuscript. The authors have adequately addressed all reviewer comments, and the manuscript fulfills all the requirements for publication in PLOS Digital Health